# Coreset Clustering on Small Quantum Computers

Teague Tomesh [1,2,*], Pranav Gokhale [2], Eric R. Anschuetz [3] and Frederic T. Chong [2,4]

1 Department of Computer Science, Princeton University, Princeton, NJ 08540, USA
2 Super.tech, Chicago, IL 60615, USA; pranav@super.tech (P.G.); chong@cs.uchicago.edu (F.T.C.)
3 MIT Center for Theoretical Physics, Massachusetts Institute of Technology, Cambridge, MA 02139, USA; eans@mit.edu
4 Department of Computer Science, University of Chicago, Chicago, IL 60637, USA
* Correspondence: ttomesh@princeton.edu

**Abstract:** Many quantum algorithms for machine learning require access to classical data in superposition. However, for many natural data sets and algorithms, the overhead required to load the data set in superposition can erase any potential quantum speedup over classical algorithms. Recent work by Harrow introduces a new paradigm in hybrid quantum-classical computing to address this issue, relying on coresets to minimize the data loading overhead of quantum algorithms. We investigated using this paradigm to perform *k*-means clustering on near-term quantum computers, by casting it as a QAOA optimization instance over a small coreset. We used numerical simulations to compare the performance of this approach to classical *k*-means clustering. We were able to find data sets with which coresets work well relative to random sampling and where QAOA could potentially outperform standard *k*-means on a coreset. However, finding data sets where both coresets and QAOA work well—which is necessary for a quantum advantage over *k*-means on the entire data set—appears to be challenging.

**Keywords:** quantum computing; machine learning; QAOA

## 1. Introduction

Machine learning algorithms for analyzing and manipulating large data sets have become an integral part of today's world. Much of the rapid progress made in this area over the past decade can be attributed to the availability of large data sets for machine learning algorithms to train on, and advances in hardware, such as the graphics processing unit, which accelerate the training process. The last decade has also witnessed the emergence of prototype quantum computers which have been implemented using various qubit technologies, including superconducting circuits, trapped ions, neutral atoms, and solid state devices [1–4]. The intersection between the emergent machine learning and quantum computing fields has produced many new algorithms which promise further advances in data processing capabilities.

Quantum algorithms such as HHL for solving linear systems [5] and Grover's algorithm for database search [6] are known to achieve exponential and quadratic speedups over their classical counterparts, respectively. However, many quantum algorithms for machine learning, including HHL and Grover search, also assume the use of an input data model (e.g., quantum RAM [7]) which allows them to easily load classical data onto the quantum processor. This model is currently unrealistic [8]. Without access to quantum RAM, which presents the state $|\psi\rangle$ on demand, we resort to using a quantum circuit to generate the desired state $|\psi\rangle$. Unfortunately, as the size of classical data sets grows to millions or billions of data points, the time and space requirements necessary to load the data may erase any potential quantum speedup.

Recent work by Harrow [9] introduced a new paradigm of hybrid quantum-classical computing to address this issue. The main idea is to take a large classical data set *X* and

use a classical computer, potentially aided by a small quantum processor, to construct a coreset: a smaller data set $X'$ combined with a weight function $w : X' \rightarrow \mathbb{R}_{\geq 0}$ which sufficiently summarizes the original data. If the coreset is small enough (but still a faithful representation of $X$), one could hope to optimize under the coreset with a small quantum computer. Prior work has focused on finding coreset construction algorithms that allow machine learning models to train on the coreset while remaining competitive with models that are trained on the entire data set [10–12].

In [9], Harrow proposes three new hybrid algorithms which cover a range of machine learning tasks including maximum a posteriori estimation, inference, and optimization. We evaluate the first of Harrow's new algorithms and adapt it to noisy quantum computers. The general version of this algorithm takes a data set $X$ and cost function $f$ as input, uses a classical computer to construct a coreset $(X', w)$, and then uses a quantum optimization algorithm to perform maximum a posteriori estimation ([9], Algorithm 1). A specific instance of the algorithm is also outlined which solves the $k$-means clustering problem ([9], Algorithm 1.1). The specific case of $k$-means is the focus of this paper. At a high level, this algorithm solves $k$-means clustering on a data set $X$ by first constructing a coreset $(X', w)$, and then optimally clustering $(X', w)$ with Grover search.

However, Grover search is unlikely to be tenable on noisy, near-term devices [13]. As proposed in [9], we reformulated the coreset clustering problem as a quantum approximate optimization algorithm (QAOA) [14] instance. QAOA is variationally optimized and is able to tolerate some noise when coupled with a robust classical optimizer. For simplicity, we restricted our study to 2-means clustering problems, and Algorithm 1 summarizes our approach.

---

**Algorithm 1:** 2-means clustering via coresets+QAOA.

| | |
|---|---|
| **Input** | : A data set $x_1, ..., x_n \in \mathbb{R}^d$ |
| **Output** | : Cluster centers $\mu_{-1}$ and $\mu_{+1}$ which approximately minimize |

$$\sum_{i \in [n]} \min_{j \in \{-1, +1\}} \left\| x_i - \mu_j \right\|^2$$

**Algorithm:**
1. Construct a coreset $(X', w)$ of size $m$.
2. Construct a $j$th order $m$-qubit Hamiltonian for the coreset.
3. Use QAOA to variationally approximate an energy-maximizing eigenstate of the Hamiltonian.
4. Treat the 0/1 assignment of the eigenstate as the $k = 2$ clustering.

---

Our contributions are as follows:

- We implemented algorithms for coresets and evaluated their performance on real data sets.
- We cast coreset clustering to a Hamiltonian optimization problem that can be solved with QAOA, and herein we demonstrate how to break past the assumption of equal cluster weights.
- We benchmarked the performance of Algorithm 1 across six different data sets, including real and synthetic data, comparing the 2-means clusterings found by quantum and classical means. We found that some data sets are better suited to coreset summarization than others, which can play a large role in the quality of the clustering solutions.

In our evaluations, the sizes of the coresets constructed in step 1 of Algorithm 1 are limited by the size of the quantum computer used in step 3. For some data sets, this restriction on coreset size negatively impacts the performance of clustering on the coresets when compared to $k$-means on the entire data set. Nonetheless, we were able to find cases where QAOA-based clustering on the coresets is competitive with the standard 2-means algorithms on the same coresets. This suggests that the performance of Algorithm 1 will

improve as quantum processors add additional high-quality qubits, thereby allowing them to utilize larger coresets. However, our evaluations also suggest that either large $m$ (i.e., many qubits) or a high order QAOA implementation (i.e., many gates) will be needed for a possible quantum advantage on typical data sets.

The rest of the paper is organized as follows. In Section 2 we give an overview of the $k$-means clustering problem. Section 3 discusses coresets for $k$-means. Section 4 describes the reduction from $k$-means to QAOA. Finally, we present and discuss our results in Sections 5 and 6.

## 2. $k$-Means Clustering

The $k$-means clustering problem takes an input data set $x_1, \ldots, x_n \in \mathbb{R}^d$ and aims to identify cluster centers $\mu_1, \ldots, \mu_k$ that are near the input data. Typically, $k \ll n$; for simplicity we focus on $k = 2$. Foreshadowing quantum notation, we will prefer to denote our cluster centers as $\mu_{-1}$ and $\mu_{+1}$. Then, the objective of this 2-means problem is to find the partitioning of $[n]$ into two sets $S_{-1}$ and $S_{+1}$ that minimizes the squared-distances from the closest cluster centers:

$$C(\mu_{\pm 1}) = \sum_{i \in S_{-1}} \|x_i - \mu_{-1}\|^2 + \sum_{i \in S_{+1}} \|x_i - \mu_{+1}\|^2. \tag{1}$$

While the cluster centers $\mu_{-1}$ and $\mu_{+1}$ appear to be free variables, it can be shown [15] that they are uniquely determined by the $S_{-1}$ and $S_{+1}$ partitionings that minimize Equation (1). In particular, these cluster centers are the centroids:

$$\mu_{-1} = \frac{\sum_{i \in S_{-1}} x_i}{|S_{-1}|} \quad \text{and} \quad \mu_{+1} = \frac{\sum_{i \in S_{+1}} x_i}{|S_{+1}|}.$$

Thus, in principle the objective function in Equation (1) can be minimized by evaluating all $2^n$ possible partitionings of $[n]$ into $S_{-1}$ and $S_{+1}$. However, this brute force exponential scaling is impractical, even for modest $n$. Instead, $k$-means is typically approached with heuristics such as Lloyd's algorithm [16], which does not guarantee optimal solutions in polynomial time, but performs well in practice. Moreover, relatively simple improvements to the initialization step in Lloyd's algorithm leads to performance guarantees. Notably, the $k$-means++ initialization procedure guarantees $(8 \ln k + 2)$-competitive solutions in the worst case [17].

For many data sets, Lloyd's algorithm augmented with $k$-means++ initialization rapidly converges to close-to-optimal (often optimal) solutions. However, in general, finding the optimal cluster centers is a computationally hard problem. Even for $k = 2$, the problem is NP-hard [18].

## 3. Coresets for $k$-Means

An $\epsilon$-coreset for $k$-means is a set of $m$ (typically $\ll n$) weighted points such that the optimal $k$-means clustering on the coreset, $\tilde{C}^*$, is within $(1 + \epsilon)$ of the optimal clustering on the entire data set of $n$ points, $C^*$. In other words $\tilde{C}^* - C^* \leq \epsilon$. A coreset data reduction is appealing because we would prefer to solve a problem with $m \ll n$ points. The size $m$ needed depends on $k$, the target error $\epsilon$, the data set dimension $d$, and the probability of success $\delta$. We implemented two coreset procedures. The first, BLK17 ([10], Algorithm 2), gives a coreset size of $m = O(\frac{dk^3 \log k + k^2 \log \frac{1}{\delta}}{\epsilon^2})$. The second, BFL16 ([19], Algorithm 2), gives a coreset size of $m = O(\epsilon^{-2} k \log k \min(\frac{k}{\epsilon}, d))$.

One might hope to pick a target $\epsilon$ and then pick $m$ accordingly. However, the exact expressions—including constants—for the scaling of $m$ are not readily available. Regardless, our goal was simply to probe the limits of small current-generation quantum computers, which have at most a few dozen qubits. Therefore, we approached coreset construction in the reverse direction by first choosing $m$ and then evaluating the performance of the

resulting coreset. As discussed in the next section, $m$ will equal the number of qubits we need. Therefore, we chose $m \in \{5, 10, 15, 20\}$ for our evaluations.

For implementations of the BLK17 and BFL16 coreset algorithms, an $(\alpha, \beta)$ bicriterion approximation is required. We used $D^2$ sampling, which is the initialization step for $k$-means++ [17], as our bicriterion approximation. We chose $\beta = 2$, which corresponds to picking $\beta k = 4$ centroids in the bicriterion approximation. For each data set, we selected the best (lowest cost) approximation from 10 repeated trials, as is also done by Scikit-learn's default implementation of $k$-means.

During our evaluations, we did not find significant differences between the performance of BLK17 and BFL16. In fact, we did not observe a significant improvement over random sampling either, except for a synthetic data set with a few rare and distant clusters.

## 4. Coreset $k$-Means via QAOA

### 4.1. QAOA

The quantum approximate optimization algorithm (QAOA) [14] is a quantum variational algorithm inspired by the quantum adiabatic algorithm [20]. The adiabatic theorem implies that, for a large enough $T$, starting in the $|+\rangle^{\otimes m}$ state and performing time-evolution under the time dependent Hamiltonian:

$$H(t) = \left(1 - \frac{t}{T}\right) \sum_{i=1}^{m} X_i + \frac{t}{T} H_P$$

from $t = 0$ to $t = T$ results in a state with high overlap with the $m$ qubit state $|z_{\text{sol}}\rangle$, where $|z_{\text{sol}}\rangle = \arg\max_{|z\rangle} \langle z|H_P|z\rangle$. For concreteness we assume that $H_P$ is diagonal such that $|z_{\text{sol}}\rangle$ is a computational basis state. One can approximate this adiabatic evolution with a finite Trotterized evolution

$$|z_{\text{sol}}\rangle \approx |\boldsymbol{\beta}, \boldsymbol{\gamma}\rangle \equiv \prod_{j=1}^{p} e^{-i\beta_j H_M} e^{-i\gamma_j H_P} |+\rangle^{\otimes m} \tag{2}$$

for certain $\boldsymbol{\beta}, \boldsymbol{\gamma}$, where $H_M = \sum_{i=1}^{m} X_i$. In the limit $p \to \infty$, the Trotter decomposition and the adiabatic theorem imply that there exist $\boldsymbol{\beta}$ and $\boldsymbol{\gamma}$ such that this approximation is exact; a priori, however, it is not obvious what one should choose for these parameters, for finite $p$, to tighten the approximation in Equation (2). Therefore, QAOA combines the ansatz of Equation (2) with a classical optimization loop, performing the maximization of the function $F(|\boldsymbol{\beta}, \boldsymbol{\gamma}\rangle) = \langle \boldsymbol{\beta}, \boldsymbol{\gamma}|H_P|\boldsymbol{\beta}, \boldsymbol{\gamma}\rangle$.

By the variational principle, for large enough $p$ the $\arg\max$ of this optimization will approximate $|z_{\text{sol}}\rangle$. In practice, a quantum computer evaluates $F(|\boldsymbol{\beta}, \boldsymbol{\gamma}\rangle)$ (or, e.g., gradients of $F(|\boldsymbol{\beta}, \boldsymbol{\gamma}\rangle)$), whilst a classical computer uses the function evaluations to heuristically optimize the function. In the remainder of this section we describe how one can interpret the solution of the $k$-means problem as the highest excited state of a diagonal Hamiltonian, which can be heuristically found using QAOA. Prior work [21] also proposed and experimentally demonstrated clustering via QAOA; however, our work explicitly derives the cost functions for the equally and unequally weighted cases of $k$-means and targets *fully connected* Max-Cut instances with the use of coresets.

### 4.2. Hamiltonians for k-Means Clustering: Equal Cluster Weights

Under the weighted vectors of a coreset of size $m$, the 2-means objective function is similar to that of Equation (1), but now each input vector $\boldsymbol{x}_i$ has an associated weight $w_i$. The modified objective function is then

$$\widetilde{C}(\boldsymbol{\mu}_{\pm 1}) = \sum_{i \in S_{-1}} w_i \|\boldsymbol{x}_i - \boldsymbol{\mu}_{-1}\|^2 + \sum_{i \in S_{+1}} w_i \|\boldsymbol{x}_i - \boldsymbol{\mu}_{+1}\|^2, \tag{3}$$

where the cluster centers are now also weighted such that:

$$\boldsymbol{\mu}_{-1} = \frac{\sum_{i \in S_{-1}} w_i \boldsymbol{x}_i}{W_{-1}} \quad \text{and} \quad \boldsymbol{\mu}_{+1} = \frac{\sum_{i \in S_{+1}} w_i \boldsymbol{x}_i}{W_{+1}}.$$

Here, $W_{\pm 1} = \sum_{i \in S_{\pm 1}} w_i$. We also define $W \equiv W_{-1} + W_{+1}$.

It can be shown that minimizing Equation (3) is equivalent to maximizing the weighted intercluster distance:

$$D = W_{+1} W_{-1} \|\boldsymbol{\mu}_{+1} - \boldsymbol{\mu}_{-1}\|^2. \tag{4}$$

In this section, we consider the case where the optimal clusters have equal weights, $W_{+1} = W_{-1}$. Often, this is a good approximation, because $W_{+1} W_{-1}$ is maximized for $W_{+1} = W_{-1} = W/2$.

Under this constraint, Equation (4) reduces to

$$D = \sum_i w_i^2 \|\boldsymbol{x}_i\|^2 + 2 \sum_{i<j} w_i w_j \boldsymbol{x}_i \cdot \boldsymbol{x}_j - 4 \sum_{i \in S_{-1}, j \in S_{+1}} w_i w_j \boldsymbol{x}_i \cdot \boldsymbol{x}_j. \tag{5}$$

In this expression, only the third term is dependent on our $S_{-1}, S_{+1}$ partitioning. Therefore, the 2-means objective for equal cluster weights is equivalent to maximizing the (re-scaled) third term:

$$\widetilde{D} = \sum_{i \in S_{-1}, j \in S_{+1}} -w_i w_j \boldsymbol{x}_i \cdot \boldsymbol{x}_j. \tag{6}$$

Equation (6) can be interpreted as a weighted Max-Cut problem on a complete graph. Each vertex in the graph represents a coreset point $\boldsymbol{x}_i$, and the weight of the $(i, j)$ edge is $-w_i w_j \boldsymbol{x}_i \cdot \boldsymbol{x}_j$. Our objective was to assign labels $\pm 1$ to each vertex in the graph such that the sum of edge weights over cut-crossing edges is maximized. Figure 1 depicts an example for a coreset containing five points. Although all 10 edges have weights, we only summed the weights of edges crossing the cut. For the particular coloring (partitioning) in Figure 1, six of the edges cross the cut.

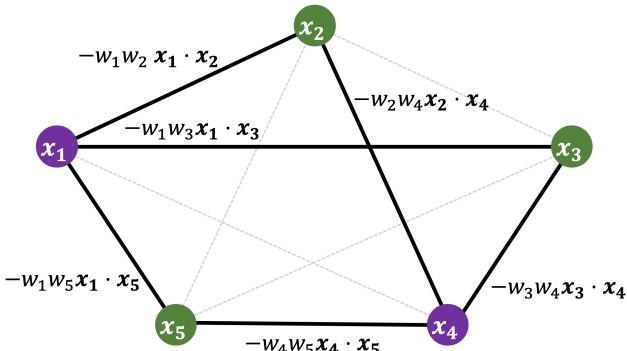

**Figure 1.** Weighted Max-Cut for a coreset consisting of five points. Given an assignment of vertices to two colors, i.e., a cut, we are interested in the sum of $-w_i w_j \boldsymbol{x}_i \cdot \boldsymbol{x}_j$ on edges crossing the cut. By interpreting these terms as edge weights, we seek a weighted Max-Cut.

In order to maximize Equation (6) using QAOA, we must encode it as a Hamiltonian. For $Z_i, Z_j \in \{-1, +1\}$, note that $\frac{1 - Z_i Z_j}{2}$ is 1 for $Z_i \neq Z_j$ and 0 for $Z_i = Z_j$. Therefore, Equation (6) corresponds to the energy of the problem Hamiltonian:

$$H_P = \frac{1}{2} \sum_{i<j} w_i w_j \boldsymbol{x}_i \cdot \boldsymbol{x}_j (Z_i Z_j - 1). \tag{7}$$

### 4.3. Hamiltonians for k-Means: Unequal Cluster Weights

Now, we once again begin with Equation (4). Unlike in Section 4.2, however, we no longer assume that $W_{+1} = W_{-1} = W/2$. We can instead write:

$$
\begin{aligned}
W_{-1}W_{+1}\|\boldsymbol{\mu}_{-1} - \boldsymbol{\mu}_{+1}\|^2 &= \left\|\frac{\sqrt{W_{-1}W_{+1}}}{W_{-1}}\sum_{i\in S_{-1}}w_i\boldsymbol{x}_i - \frac{\sqrt{W_{-1}W_{+1}}}{W_{+1}}\sum_{i\in S_{+1}}w_i\boldsymbol{x}_i\right\|^2 \\
&= \left\|\frac{\sqrt{W_{+1}}}{\sqrt{W_{-1}}}\sum_{i\in S_{-1}}w_i\boldsymbol{x}_i - \frac{\sqrt{W_{-1}}}{\sqrt{W_{+1}}}\sum_{i\in S_{+1}}w_i\boldsymbol{x}_i\right\|^2 = \sum_i\left(\frac{W_{+1}}{W_{-1}} \text{ if } i\in S_{-1}, \text{ else } \frac{W_{-1}}{W_{+1}}\right)w_i^2\|\boldsymbol{x}_i\|^2 \\
&+ 2\sum_{i<j}\left(\frac{W_{+1}}{W_{-1}} \text{ if } i,j\in S_{-1}, \frac{W_{-1}}{W_{+1}} \text{ if } i,j\in S_{+1}, \text{ else} -1\right)w_iw_j\boldsymbol{x}_i\cdot\boldsymbol{x}_j.
\end{aligned}
\tag{8}
$$

By examining the ratios

$$
\frac{W_{+1}}{W_{-1}} = \frac{1}{W_{-1}/W} - 1 \quad\text{and}\quad \frac{W_{-1}}{W_{+1}} = \frac{1}{W_{+1}/W} - 1 \tag{9}
$$

we consider the Taylor expansion of these terms around the point $x \equiv W_{-1}/W = W_{+1}/W = 1/2$, i.e., around the equal cluster weight scenario of Section 4.2. The motivation for using a Taylor expansion is that the resulting polynomial has operational significance as a Hamiltonian and can be written in terms of $Z_i$'s. The function $1/x$ and the three leading orders of its Taylor expansion are shown in Figure 2.

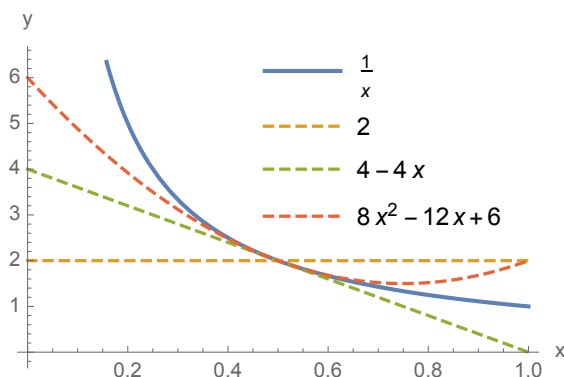

**Figure 2.** The first three Taylor expansions of $y = \frac{1}{x}$ around $x = 0.5$.

The zeroth order Taylor expansion, $1/x \approx 2$, corresponds to the case of equal cluster weights. Keeping terms through first order gives $4 - 4x$. Informally, this linear approximation is close to $1/x$ for $0.4 < x < 0.6$. Therefore, we could reasonably hope for the first order Taylor expansion to work well for slightly unbalanced optimal clusters with 3:2 imbalances.

With this motivation, we can expand Equation (9) and find that:

$$
\frac{W_{+1}}{W_{-1}} \approx 3 - \frac{4}{W}W_{-1} \quad\text{and}\quad \frac{W_{-1}}{W_{+1}} \approx 3 - \frac{4}{W}W_{+1}.
$$

By performing a similar translation into Pauli $Z$ operators as in Section 4.2, these terms can be expressed in Hamiltonian form and simplified to

$$
\frac{W_{+1}}{W_{-1}} \approx 1 + \frac{2}{W}\sum_l w_l Z_l \quad\text{and}\quad \frac{W_{-1}}{W_{+1}} \approx 1 - \frac{2}{W}\sum_l w_l Z_l.
$$

Plugging these into Equation (8) gives

$$W_{-1}W_{+1}\|\boldsymbol{\mu}_{-1}-\boldsymbol{\mu}_{+1}\|^2 = \sum_i \left(1 + \frac{2}{W}\sum_l w_l Z_l \text{ if } i \in S_{-1}, \text{ else } 1 - \frac{2}{W}\sum_l w_l Z_l\right) w_i^2 \|\boldsymbol{x}_i\|^2$$

$$+ 2\sum_{i<j}\left(1 + \frac{2}{W}\sum_l w_l Z_l \text{ if } i,j \in S_{-1}, 1 - \frac{2}{W}\sum_l w_l Z_l \text{ if } i,j \in S_{+1}, \text{ else} - 1\right) w_i w_j \boldsymbol{x}_i \cdot \boldsymbol{x}_j.$$

The indicator functions can also be rewritten as Pauli expressions, resulting in the final problem Hamiltonian:

$$\sum_i \left(1 - \frac{2Z_i}{W}\sum_l w_l Z_l\right) w_i^2 \|\boldsymbol{x}_i\|^2 + 2\sum_{i<j}\left(Z_i Z_j - \frac{Z_i + Z_j}{W}\sum_l w_l Z_l\right) w_i w_j \boldsymbol{x}_i \cdot \boldsymbol{x}_j. \tag{10}$$

Interestingly, accounting for unequal cluster weights produces a problem Hamiltonian (Equaton (10)) containing only quadratic terms. Therefore, implementing this Hamiltonian on a quantum computer as part of QAOA would require only single and two-body interactions, i.e., no more difficult to implement than the zeroth order case that assumes equal cluster weights. However, for higher-order Taylor expansions, the degree of the Hamiltonian will increase. A second order Taylor expansion will have cubic terms, a third order Taylor expansion will have quartic terms, and so forth. These higher order terms are implemented via multi-qubit operations whose decomposition into single and two-qubit gates can quickly exhaust available quantum resources.

## 5. Results

### 5.1. Data Sets

Table 1 describes the six data sets used in our evaluations. The Epilepsy, Pulsars, and Yeast data sets are parts of the UCI Machine Learning Repository [22]. For Common Objects in Context (COCO), the image pixels were preprocessed with the img2vec [23] library. This library translates the pixels of each image into a 512-dimensional feature vector using a Resnet-18 model [24] pretrained on the ImageNet data set [25].

### 5.2. Evaluation Methodology

We evaluated 2-means on each of these data sets using four different approaches. The first three evaluation modes use classical $k$-means to find a clustering over the entire data set, a random coreset, and a coreset generated with BFL16. The final evaluation mode finds a clustering via QAOA (Algorithm 1) over the same BFL16 coreset. For classical clustering, we used Scikit-learn's [26] default implementation of $k$-means, which initializes clusters with the best of 10 $k$-means++ [17] and terminates either after 300 iterations or upon stabilizing within $10^{-4}$ relative tolerance. This default implementation is an aspirational, though realistic, target against which to compare QAOA. The cost function we compute is the "sum of squared distances to nearest cluster" objective function in Equation (3), also referred to as inertia [27].

In all of our evaluations the lowest clustering cost was found by evaluating $k$-means over the entire data set. We used this fact to rescale the coreset clustering costs, dividing their scores by the lower bound achieved with $k$-means over the full data set. This rescaling lets us better visualize the differences in performance between the different coresets and clustering methods.

On each data set, we computed $m = 5, 10, 15,$ and 20 uniformly random samples, and $m$-coresets using BFL16. Then, we ran the 2-means clustering implementation on each coreset, and evaluated the cost of the output solution against the *entire* data set. For each data set, we ran this process 10 times. For five of the six data sets, we report the best of 10 results, since in practice one would indeed choose the best result. For the synthetic data set, we report average, best, and worst costs, to emphasize that the coreset algorithm is consistently better than random sampling.

**Table 1.** Data sets evaluated.

| Data Set | Description |
| --- | --- |
| CIFAR-10 | 10,000 images ($32 \times 32$ pixels) from CIFAR-10 data set [28]. 1000 images per category. |
| COCO | 5000 images from Common Objects in Context validation data set [29]. Images translated into feature vectors of dimension 512. |
| Epilepsy | Epileptic seizure recognition data set from [30]. 11,500 vectors of dimension 179. |
| Pulsars | Pulsar candidates from HTRU2 data set [31]. 1600/17,900 of 9-dimensional feature vectors are pulsars. |
| Yeast | Localization sites of proteins [32]. 1500 8-dimensional vectors. |
| Synthetic | 40,000 512-dimensional points drawn from 11 random Gaussian clusters. Ten clusters contribute 5 points each, last cluster has majority. |

In addition to these classical results, we took the best $m = 5$ and $m = 10$ BFL16 coresets and constructed Hamiltonians for them, as described in Section 4. For $m = 10$, we constructed Hamiltonians with zeroth, first, second, and infinite order Taylor expansions. For $m = 5$, we only constructed the zeroth order Hamiltonian (i.e., assuming equal cluster weights as in Section 4.2), because this is a realistic experimental target for current devices (see the evaluation in Section 5.4).

For each Hamiltonian, we found its highest-energy eigenstate by brute force search over the $2^m$ basis states; for larger instances where brute force searching is impossible, one would approximately optimize with QAOA. This is the solution one would expect to find with Grover's search algorithm, and it can also be interpreted as a bound on the best possible QAOA performance. The highest eigenstate is the weighted Max-Cut solution, or equivalently, the best cluster assignment on the coreset. For the infinite order Hamiltonian, this highest eigenstate is truly the optimal clustering of the coreset. However, note that the optimal clustering on a coreset does not necessarily correspond to the optimal clustering on the whole data set. This is because the coreset is only an approximation of the underlying data set.

*5.3. Coreset and QAOA Bound Results*

Figure 3 shows the results of using the methodology above on our six data sets. The green and orange bars, which are entirely classical, correspond to 2-means on the random and BFL16 coresets, respectively (note that all of the costs are scaled with respect to *k*-means over the full data set). Interestingly, for the majority of our benchmarked data sets, the BFL16 and random coresets show similar performances. The only data set whereon BFL16 consistently outperformed the random coresets was the synthetic data set, which has 39,950 points in a localized Gaussian cluster, along with 50 points in ten distant clusters. On this data set, random sampling did not perform well, because the random samples were unlikely to capture points outside of the big cluster. This suggests we may see gains from using coresets on data sets oriented towards anomaly-detection.

The blue bars in Figure 3 correspond to the energy-maximizing eigenstate of each Hamiltonian, which was constructed from a BFL16 coreset of $m$ elements and a given Taylor expansion order (indicated by the tick labels). QAOA attempts to find this energy-maximizing eigenstate, but it is only an approximate solver. Therefore, the blue bars can be interpreted as a lower bound on the cost of a QAOA-based solution. However, the approximate nature of QAOA could serve as a benefit to the clustering algorithm overall. Recall that the final clustering cost is evaluated with respect to the whole data set, and the coreset is only an approximation of this data set. Therefore, a suboptimal eigenstate of the Hamiltonian (i.e., a suboptimal clustering of the coreset) can actually outperform the

solution obtained via the Hamiltonian's optimal eigenstate because the cost is evaluated over the full data set.

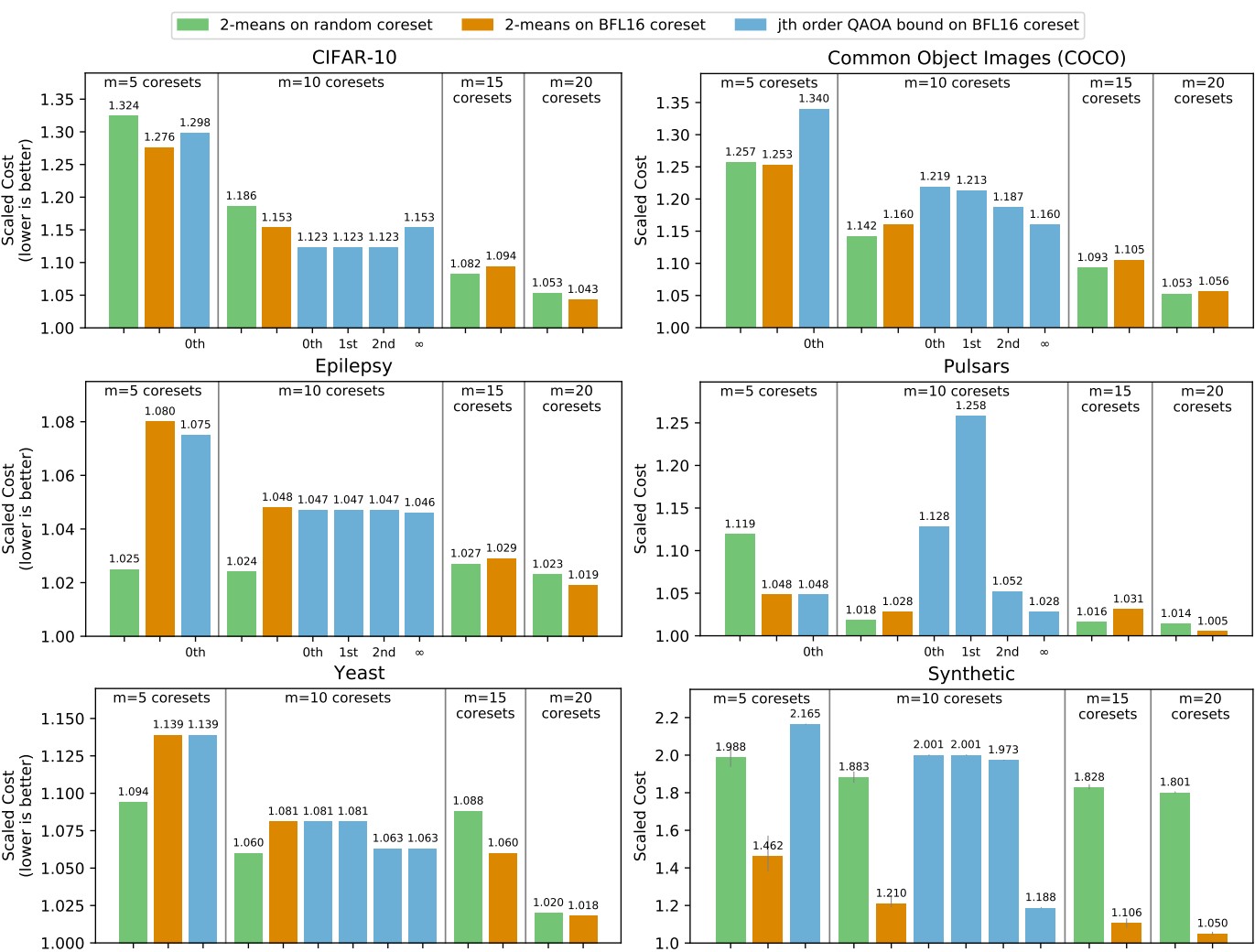

**Figure 3.** Evaluation of quantum and classical coreset clustering on six different data sets. The green and orange bars were obtained by running classical 2-means on $m = 5, 10, 15,$ and 20 random and BFL16 coresets, respectively. The blue bars express the cost of the highest-eigenstate of the $m = 5$ and 10 Hamiltonians using a $j$-th order Taylor expansion. These can be interpreted as bounds on QAOA's performance with $m$ qubits. We report the best of 10 results for all data sets, except for the synthetic one, which shows the means and min-max error bars. All costs are scaled with respect to the cost achieved by running 2-means over the full data set.

We observed this behavior in a few data sets. Focusing on the blue bars only, the cost was almost always lower (or the same) when we increased the order of the Hamiltonian (see Figure 2). However, for the CIFAR-10 and Pulsars data sets, there were lower-order Taylor expansions (i.e. more approximate Hamiltonians) of the $m = 10$ coresets, which had lower costs than the higher-order expansions. We also found cases where a coarser approximation to the data set provides better clustering than a low-order expansion of a larger coreset. In the Pulsars experiment, the optimal clustering of the $m = 10$ coreset found by the $\infty$-order QAOA bound and the classical 2-means (Scaled Cost = 1.028) does outperform the coarser $m = 5$ coreset clustering. However, the zeroth order QAOA bound on this coarse $m = 5$ coreset (Scaled Cost = 1.048) is able to outperform the zeroth, first, and second order approximations of the $m = 10$ coreset. These examples highlight the interplay between the two levels of approximation within this algorithm: the coreset approximation of the data set and the QAOA approximation of the clustering solution. It may indicate potential

tradeoffs that can be made between the classical (coreset) and quantum (QAOA) aspects of this algorithm when one or the other may be resource constrained.

On all six data sets studied, the lowest cost was achieved by running 2-means on the whole data set. The main barrier to a possible "quantum advantage" seems to be that for small $m$, the coreset approximation is too coarse. However, we were able to find QAOA bounds that outperform the corresponding classical results on the same coreset. This occurred on the CIFAR-10, Epilepsy, Yeast, and synthetic data sets, where the QAOA bound outperformed 2-means on the same BFL16 coreset. It is also encouraging that the zeroth and first order QAOA bounds, which have only quadratic gate count, are generally similar to the higher order bounds. The exceptions, where unequal cluster sizes are favored, appear in the synthetic (unbalanced by construction) and Pulsars (only a fraction of the data vectors are truly pulsars) data sets. The broader pattern appears to be that if a data set is amenable to coresets, it does not work well for low-order QAOA approximation, and vice versa.

### 5.4. Experimental QAOA Results

For each of the six data sets examined above, we also tested the performance of QAOA for the $m = 5$ case on the 5-qubit *ibmq_rome* processor (IBM, Yorktown Heights, NY, USA). Figure 4 shows the QAOA circuit. Solving $k$-means via QAOA on the zeroth or first order Hamiltonian is equivalent to finding the weighted Max-Cut on a complete graph, which requires an interaction between every pair of qubits during each step of QAOA. This would seem to imply that the quantum computer must support a high degree of connectivity between qubits or else incur a large number of SWAP operations which would deteriorate performance. However, using the swap networks proposed in [33,34], we can achieve the required all-to-all interactions in depth which scales linearly with the number of qubits $m$, while only requiring nearest-neighbor interactions. This is critical for the *ibmq_rome* processor which has a linear communication topology. Swap networks were also implemented for Max-Cut QAOA in [35] where the total number of controlled-NOT gates (CNOTs) required for $p$ layers of QAOA on $m$ qubits was found to scale as $\frac{3}{2}m(m-1)p$. We note that this CNOT count can be slightly reduced by forgoing the last layer of the SWAP network and reordering the bits in a classical post-processing step so that the overall number of CNOTs scales as $\left(\frac{3}{2}m(m-1) - \lfloor\frac{m}{2}\rfloor\right)p$.

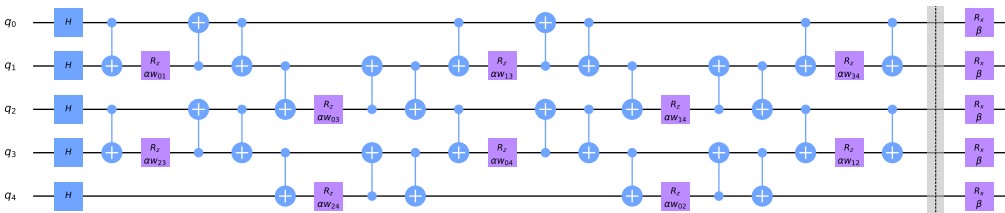

**Figure 4.** An example of a QAOA circuit used to solve the weighted Max-Cut problem on an $m = 5$ coreset implemented using the swap networks proposed in [33,34]. Here, $p = 1$, $\alpha$, and $\beta$ are the variational parameters and the $w_{ij}$'s are the edge weights of the constructed graph (see Figure 1). Using the swap network, each qubit is able to interact with every other qubit, $O(n^2)$ interactions, in linear depth.

Figure 5 shows the results of running $k$-means QAOA for the Epilepsy $m = 5$ coreset on *ibmq_rome* with and without the swap network. Before running on the quantum hardware, we found optimal values for the variational parameters using noisy circuit simulation in conjunction with a Nelder–Mead optimizer. In Figure 5 the best partitionings found by the noiseless simulation are the symmetric 01100 and 10011 states. We used the cluster centers given by the coreset partitioning to evaluate the cost function on the entire data set; both the 01100 and 10011 partitions had a cost of $c = 5.581 \times 10^{10}$. Dividing this value by the cost achieved with $k$-means over the full data set ($c_{full} = 5.419 \times 10^{10}$) gives a rescaled cost of $\frac{c}{c_{full}} = 1.030$, which approaches the bound of 1.0 and outperforms the other $m = 5$

and 10 BFL16 coresets for Epilepsy shown in Figure 3. There is a significant amount of noise present in the hardware executions compared to the noiseless simulation, but when the swap network was utilized, the 01100 solution state was still the most likely state to be measured. Without the swap network, the highest probability state would result in a suboptimal partition.

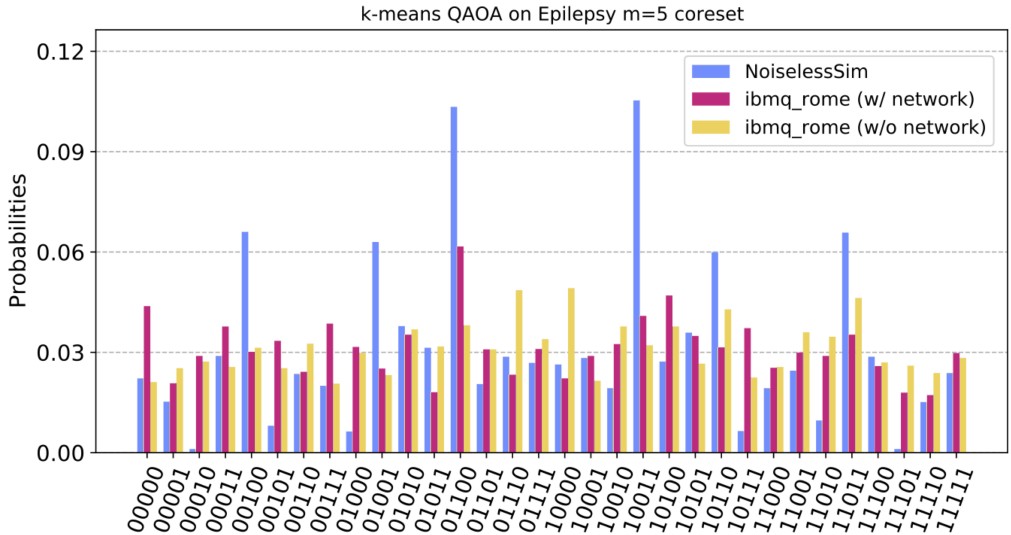

**Figure 5.** An experimental evaluation of a QAOA circuit implemented with and without the swap network [33,34]. Each distribution consists of 8192 individual shots. The noisy execution of the quantum hardware becomes apparent when comparing the experimental results with the noiseless simulation. However, by utilizing the swap network, one of the optimal bitstrings (01100) can still be identified in the output distribution with high probability.

## 6. Discussion and Conclusions

In this work we investigated the performance of *k*-means clustering via QAOA, using offline coresets to effectively reduce the size of the target data sets. Indeed, there do exist data sets where coresets seem to work well in practice, such as the synthetic data set we analyzed in Section 5.3. Furthermore, as our Hamiltonian construction of the problem is all-to-all connected, our QAOA instance circumvents the light cone oriented issues [36] associated with running constant *p*-depth QAOA on sparse bounded-degree graphs.

However, in practice, coresets constructed via BFL16 and random sampling had similar performances on the standard classification data sets we benchmarked. This may have been due to the small *m* we restricted our coresets to, with the motivation of fitting the problem instance to today's quantum computers. Alternatively, it may have been due to the fact that these "natural" data sets have near equally sized optimal clusters. Indeed, the synthetic data set where coresets performed well had artificially rare clusters that naive random sampling would miss—however, this worked to the detriment of our Hamiltonian construction of the problem, which involves Taylor expanding the optimization problem around near equal optimal cluster sizes. As standard *k*-means already performs remarkably well, it seems that one would need a high-degree Hamiltonian expansion for a method such as QAOA to compete, or otherwise a more clever Hamiltonian construction of the problem. Methods such as Grover's algorithm, however, would not necessarily have this limitation. We leave for future work refining this intuition and perhaps finding instances where both coresets and QAOA work well in conjunction.

**Author Contributions:** Conceptualization, T.T., P.G., E.R.A. and F.T.C.; methodology, T.T. and P.G.; software, T.T. and P.G.; writing—original draft preparation, T.T., P.G. and E.R.A.; writing—review and editing, T.T., P.G., E.R.A. and F.T.C.; funding acquisition, F.T.C. All authors have read and agreed to the published version of the manuscript.

**Funding:** This work is funded in part by EPiQC, an NSF Expedition in Computing, under grants CCF-1730082/1730449; in part by STAQ under grant NSF Phy-1818914; in part by DOE grants DE-SC0020289 and DE-SC0020331; and in part by NSF OMA-2016136 and the Q-NEXT DOE NQI Center. This material is based upon work supported by the Air Force Office of Scientific Research under award number FA9550-21-1-0033 and the National Science Foundation under Grant No. 2110860. E.A. is supported by the National Science Foundation Graduate Research Fellowship Program under grant number 4000063445, and a Lester Wolfe Fellowship and the Henry W. Kendall Fellowship Fund from M.I.T.

**Data Availability Statement:** The data sets used in this work are publicly available. The code is available online in our Github repository at https://github.com/teaguetomesh/coresets, accessed on 20 April 2020.

**Conflicts of Interest:** Fred Chong is Chief Scientist at Super.tech and an advisor to Quantum Circuits, Inc.

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
