# Peer review of "Coreset Clustering on Small Quantum Computers"

_electronics, doi:10.3390/electronics10141690_

Round 1

Reviewer 1 Report

This work analyzes the effect of using coreset for clustering tasks, which would be a key ingredient to apply small-scale quantum computers for machine learning. They restrict the number of elements in coreset to be <20 to ensure that we can apply current quantum computers for this purpose. They also derived a QUBO formulation of k-means clustering.

I think this work can be a solid contribution to the field. I did not find any scientific flaw in this manuscript, and hence I can recommend publication in its present form.

I spotted a typo in line 157: we now longer -> we no longer?

Author Response

Thank you for your review, and for pointing out the typo. We have updated the text to correct this error.

Reviewer 2 Report

It is a good paper to see performance comparison between quantum/classical computing in a hybrid computing. Before a publish, I am curious about the following questions:

  1. Describe your platform since we have no idea about your “small” quantum computers
  2. In Fig3, there is way much more information each bar graph. Please reorganize them so that reader can focus on logical thoughts instead of figuring out which is which in bar graphs. 
  3. In Fig 5, although it is apparent between w/ and w/o noisy, most data are around 0.03 which makes this comparison meaningless. Also, for the legend, light blue and blue is hard to be distinguished. 

Author Response

Thank you for your thoughtful comments and questions.

  1. "Small Quantum Computers" refers to the current Noisy Intermediate-Scale Quantum (NISQ) era of quantum computing. The current quantum devices that are available are characterized by their limited qubit counts and noisy gate operations. In our work, specifically, we consider the small, superconducting quantum computers available through IBM's cloud service. Our experimental results utilize one of their processors: the 5-qubit ibmq_rome device to demonstrate the utility of the fermionic-swap network for these QAOA applications.
  2. Thank you for pointing this out. We have reformatted Figure 3 to group together the results based on the size of the coreset, removed the hatches on the bars, and distinguished between the random and BFL16 classical results by making them green and orange, respectively.
  3. Per your suggestion, we have altered the color scheme of Figure 5 to better differentiate between the noiseless simulations and the noisy hardware execution. In response to your point that most of the data is around 3%: this is an important observation, this is why the typical mode of quantum circuit execution involves executing the same circuit many times. In our experiment the probability distributions "w/ network" and "w/o network" were built up over 8192 individual circuit executions - the result of each individual "shot" is a single 5-bit string. In this way, we are able to distinguish between the optimal solution (appearing in about 6% of the shots) and the other sub-optimal solutions that appeared with roughly 3% probability each. In addition, we note that even in the noiseless simulations, the optimal bitstrings only appear with ~10% probability each. So it is not necessarily the absolute value of the probability that is important, but rather, the relative differences between the optimal and suboptimal solutions.

Reviewer 3 Report

in this paper the authors demonstrated how to solve clustering problem on a small scale quantum computer aided by corsets obtained via classical algorithms. The QAOA algorithm is proposed for the quantum part. The theoretical contribution is modest, but the case study is interesting. To my opinion, it is important to have meaningful case studies which validate the benefit of having a noisy small-scale quantum computer.

Some types

  1. line 49. maximum a posteriori
  2. line 157, now longer

Author Response

Thank you for your review, and for spotting the typos in our manuscript. We have updated the text to correct these errors.